# Circulating exosomes express $\alpha_4\beta_7$ integrin and compete with CD4+ T cells for the binding to Vedolizumab

**Rossana Domenis**[1], **Marco Marino**[2], **Adriana Cifù**[1], **Giulia Scardino**[2],
**Francesco Curcio**[1,3], **Martina Fabris**[3]*

**1** Department of Medical Area, University of Udine, Udine, Italy, **2** Gastroenterology, Azienda Sanitaria Universitaria Integrata Friuli Centrale, Udine, Italy, **3** Clinical Pathology, Azienda Sanitaria Universitaria Integrata Friuli Centrale, Udine, Italy

* martina.fabris@asufc.sanita.fvg.it

**Data Availability Statement:** All relevant data are within the manuscript.

**Funding:** This work was supported by Grant Interreg ARTE (J22F170001005) and partially

## Abstract

Vedolizumab (VDZ) is a therapeutic monoclonal antibody approved for the treatment of inflammatory bowel diseases (IBD). VDZ selectively binds to the α4β7 integrin and blocks trafficking of a specific subset of gastrointestinal-homing T-lymphocytes to inflamed tissue. Although VDZ has shown promising results in numerous clinical studies a subgroup of patients do not respond adequately. Mechanistic insights and prognostic biomarkers able to predict which patients might benefit from VDZ therapy are currently lacking. Circulating exosomes were isolated from serum of blood donors and VDZ-treated patients by polymer-based precipitation. The surface expression of α4β7 integrin was evaluated by flow cytometry and the levels of exosome-bound VDZ were investigated by Promonitor-VDZ ELISA kit. The capacity of exosomes to interfere with the adhesion of VDZ-treated CD4+ T cells was assessed by adhesion assay. In this study, we showed that serum exosomes isolated from both blood donor and ulcerative colitis patients express on their surface the VDZ target α4β7 integrin. We observed an increased exosomal sequestration of VDZ in anti-TNF exposed patients compared to anti-TNFα naïve patients, according to a greater expression of α4β7 integrin on vesicles surface. Circulating exosomes could compete for VDZ binding with CD4+ T cells since we found that the amount of VDZ bound to T cells was impaired in the presence of exosomes. In addition, we demonstrated that exosomes bind VDZ, which consequently becomes unable to block MadCAM-1-mediated adhesion of lymphocytes. Circulating exosomes might contribute to drug sequestration, possibly affecting the therapeutic efficacy of VDZ in IBD patients. Our data suggest that previous biologic therapy may have altered the sequestration capacity of circulating exosomes, thus reducing the efficacy of VDZ in patients who failed anti-TNF agents.

supported by an unrestricted grant from VivaBioCell S.p.A. The funders had no role in study design, data collection and analysis, decision to publish, or preparation of the manuscript.

**Competing interests:** This work was partially supported by an unrestricted grant from VivaBioCell S.p.A. This does not alter our adherence to PLOS ONE policies on sharing data and materials.

## Introduction

Inflammatory bowel diseases (IBD) are a chronic and highly heterogeneous group of diseases, including Crohn's disease (CD) and ulcerative colitis (UC), which are characterized by inflammation of the intestinal tract. Although the exact etiopathogenic mechanism of IBD remains unknown, accumulating evidence suggests that they result from an inappropriate inflammatory response to different intestinal antigens in a genetically susceptible host [1].

The increased knowledge of the pathways involved in IBD pathogenesis has led to the development of new treatment options. Biologic agents, such as monoclonal antibodies against specific cytokines or receptors, represent a class of therapeutic molecules with the capacity to selectively block the major proinflammatory cascades involved in IBD and, consequently, the activation and proliferation of intestinal T lymphocytes, therefore reestablishing the balance between pro- and anti-inflammatory stimuli [2].

The infiltration of leukocytes in the gut mucosa is mediated in part by interactions between integrins expressed on lymphocytes, and their ligands MAdCAM-1 on the gut epithelial cells [3]. Vedolizumab (VDZ) is a humanized IgG1 monoclonal antibody that binds specifically to the $\alpha_4\beta_7$ integrin expressed on a subset of primed gut-homing T-lymphocytes [4]. The inhibition of lymphocytes migration selectively downregulates gut inflammation, while preserving systemic immune responses.

Although VDZ is successfully used for treatment of IBD naïve patients or for patients who have failed TNF-$\alpha$ antagonists [5], a significant proportion of patients does not respond at induction or loses response over time, and might develop significant side effects [6–8]. Primary or secondary loss of response to biological therapy may be caused by several potential mechanisms: advanced disease, inadequate dosing, development of neutralizing anti-drug antibodies (ADA), disease heterogeneity/cytokine switch, metabolism, accelerated clearance [9]. Clearly, a better understanding of the factors associated with drug response may allow early identification of patients who are most likely to benefit from VDZ or any other biologic agent [10].

Exosomes are a population of bilayer nanometer-sized extracellular vesicles that originate from the multivescicular endosome and are secreted by several types of living cells in all body fluids. They are regarded as major players in cell-cell communication as they transfer specific lipid, nucleic acids, proteins and other bioactive molecules from the parental cells to the neighboring or distant recipient cells [11]. In addition to their physiological functions, exosomes are involved in the pathological development and progression of numerous diseases. Recently, it has been proposed that they have a role in the pathogenesis of diverse inflammatory conditions, including cancer, type 2 diabetes, obesity, rheumatoid arthritis and neurodegenerative diseases, creating a microenvironment that triggers inflammation and sustains disease progression [12].

To date there are few studies examining the role of exosomes in IBD. Extracellular vesicles isolated from luminal fluid of IBD patients exhibit a proinflammatory effect on epithelial cells and macrophages, inducing the release of the pro-inflammatory cytokine IL-8 [13]. In addition, exosomes released by intestinal epithelial cells and naïve macrophages upon CD-associated adherent-invasive Escherichia coli infection can trigger and amplify proinflammatory responses [14]. Moreover, serum-derived exosomes in a murine model of induced acute colitis, transport acute-phase proteins and immunoglobulins that are mainly involved in the complement and coagulation cascade [15].

The content of exosomes is cell type specific and reflects the phenotypic state of the parental cell, hence exosomes are considered "fingerprints" of the releasing cell. Therefore, they could be potentially used as disease specific markers. For example, it has been reported that the level

of salivary exosomal proteasome subunit alpha type 7 (PSMA7) differs significantly between healthy individuals and IBD patients and may reflect early disease [16]. It has also been reported that patients with IBD secrete an increased number of annexin-1-containing exosomes in the blood stream, possibly released by intestinal epithelial cells to help restore damaged mucosa [17].

Interestingly, it has been proposed that exosomes could also be involved in mediating drug resistance by sequestration of therapeutic molecules into the vesicles. Specifically, exosomes secreted by breast cancer patients' serum, carrying HER2 antigen that is the target of Trastuzumab, contribute to drug sequestration, reducing therapeutic efficacy [18].

In this study, we characterized exosomes isolated from serum of ulcerative colitis patients treated with VDZ in order to address the possibility that circulating exosomes bind VDZ and interfere with its therapeutic efficacy.

## Methods

### Study population

This single-centre retrospective study was approved by the Institutional Review Board of the Department of Medical Area (IRB-DAME), University of Udine (10/IRB_CURCIO_2019). The study included 17 ulcerative colitis (UC) patients (mean age $40.7 \pm 17.5$ years, 50% female), 6 naïve to biologics and 11 previously treated with at least one anti-TNFα agent (Infliximab n = 2, Adalimumab n = 3 or both n = 6), starting Vedolizumab (VDZ) therapy and followed at the Gastroenterology Unit of the University Hospital of Udine. Baseline patient details including disease duration, smoking status, concurrent medications and Mayo score were collected. Demographic and disease characteristics of the patients at baseline were listed in Table 1 (failure to TNF-α) and in Table 2 (naïve to TNF-α). All the subjects at the beginning of VDZ therapy had confirmed moderately to severely active ulcerative colitis, defined as a total score of 6 to 12 on the Mayo scale (total Mayo scores range from 0 to 12, with higher scores indicating more severe disease) and a subscore between 2 or 3 on the endoscopic component of the Mayo scale. Of the 11 subjects previously treated with at least one anti-TNFα agent, 6 interrupted the treatment for primary failure, 2 for secondary failure and 3 for adverse effects. Enrolled patients had been treated with VDZ for at least 2 months (range 2 to 18 months). Peripheral blood samples were collected immediately before VDZ infusion and

**Table 1. Baseline characteristics of the IBD patients anti TNF-α failure.**

| Pt | Age/Sex | Lenght Disease (yrs) | Mayo score Clin/End | Smokers (Y/N) | TNF-α drugs | TNF-α Failure PF/SF/AE | VDZ Resp. R/PF/SF |
|---|---|---|---|---|---|---|---|
| 1 | M/49 | 5 to 10 | 12/3 | N | IFX | PF | PF |
| 2 | M/59 | > 10 | 6/3 | Y | ADL | PF | R |
| 3 | M/54 | 5 to 10 | 8/2 | N | ADL/IFX | PF | R |
| 4 | M/41 | < 5 | 9/3 | N | IFX | AE | R |
| 5 | M/31 | 5 to 10 | 7/2 | N | ADL/IFX | SF | SF |
| 6 | F/22 | < 5 | 8/2 | N | ADL | AE | SF |
| 7 | F/24 | 5 to 10 | 8/3 | N | ADL/IFX | PF | SF |
| 8 | F/36 | 5 to 10 | 7/3 | N | ADL/IFX | SF | R |
| 9 | F/31 | < 5 | 8/3 | N | ADL | PF | R |
| 10 | F/39 | < 5 | 8/2 | N | ADL/IFX | AE | R |
| 11 | M/19 | < 5 | 7/3 | N | ADL/IFX | PF | SF |

IFX = infliximab, GOL = golimumab, ADL = adalimumab, R = responder, PF = primary failure, SF = secondary failure, AE = adverse event.

**Table 2. Baseline characteristics of the IBD patients naïve to TNF-α.**

| Pt | Age/Sex | Lenght Disease (yrs) | Mayo score Clin/End | Smokers (Y/N) | VDZ Resp. R/PF/SF |
|----|---------|----------------------|---------------------|---------------|-------------------|
| 1 | M/74 | 5 to 10 | 7/3 | N | R |
| 2 | F/68 | 5 to 10 | 8/2 | N | SF |
| 3 | F/35 | < 5 | 9/3 | N | R |
| 4 | F/19 | < 5 | 8/3 | N | R |
| 5 | F/43 | > 10 | 7/2 | N | SF |
| 6 | M/23 | < 5 | 9/3 | N | R |

R = responder, PF = primary failure, SF = secondary failure.

centrifuged immediately at 2500g for 10 min to obtain the serum which was stored in aliquots at -80˚C until analysis. Sera from 12 age and sex-matched healthy adults (mean age 41 ± 16.5 years, 50% female) were used as controls. This retrospective study was approved by the local Institutional Review Board (10/IRB_CURCIO_2019).

## Exosome isolation

Exosomes were isolated from the serum of patients and blood donors (CTRL) with three different technologies. Serum (250 μL) was centrifuged at 3000g for 15 minutes, filtered with 0.22 μm filter and concentrated in Amicon® Ultra-4 (100 kDa). Samples were incubated with ExoQuick (System Biosciences) for 30 minutes at 4˚C and then centrifuged twice at 1500g for 30 and 5 minutes, respectively. The exosome-containing pellet was then resuspended in 250 μL of PBS buffer and stored -20˚C for subsequent analysis. Alternatively, 2 ml of serum was overlaid on qEV2 size exclusion column (Izon) followed by elution with PBS, according to manufacturer's protocol. The fractions contained exosomes were pooled, filtered with 0.22 μm filter and concentrated in Amicon® Ultra-4 (100 kDa) to a final volume of 150 μL. Finally, exosomes were extracted by exoEasy Maxi Kit (Qiagen), according to the manufacturer's protocol. Briefly, 4 ml of pre-filtered serum was mixed with an equal volume of buffer XBP, transferred into the exoEasy spin column and centrifuged at 500g for 1 min. The flow-through was discarded and the spin column was washed with 10 mL buffer XWP. Finally, 400 μL of buffer XE was added to membrane and spin column was centrifuged at 5000g for 5 min to elute the exosomes.

## Exosome characterization

The number of isolated exosomes was determined using the Exocet kit (System Biosciences), according to manufacturer's protocol [19].

Exosomes were analyzed for the expression of exosomal markers and proteins expressed on immune cells surface by flow cytometry and immunoblotting [20]. Specifically, vesicles ($3 \times 10^9$) purified by immunoaffinity Exo-Flow kit (System Biosciences) were stained with specific monoclonal antibodies anti-CD81 FITC (Biolegend), anti-CD63 FITC (Santa Cruz), anti-CD9 PE (eBiosciences), anti-CD3 APC (eBiosciences), anti-CD14 PE (eBiosciences), anti-MadCAM-1 FITC (LSBio) and anti-integrin $\alpha_4\beta_7$ FITC (Biorbyt). The flow cytometry analysis was performed calculating the percentage of exosome-bound beads compared with beads alone. Isolated exosomes (50 μg for exosomes isolated by Exoquick and exoEasy and 25 μg for exosomes isolated by qEv2) were boiled in Laemmli sample buffer and subjected to electrophoresis on Mini-PROTEAN TGX precast 10% gels (BioRad). Primary antibodies against flotillin-2 (1:1000, Cell Signaling), CD9-HRP conjugated (1:500, Novus biologicals) and TSG101-HRP

conjugated (1:1000, Santa Cruz) were used for immunoblotting analysis. Exosomes were analyzed by nanoparticle tracking analysis (NTA), using the NanoSight LM10 system (Malvern). For each sample, diluted in PBS (1000–10,000 times), a video was captured for 60s each with a detection threshold set at 16 (maximum).

The Promonitor®-VDZ ELISA test (Progenika Biopharma-Grifols) was used to quantify the levels of VDZ in patient's serum and exosome samples, according to protocol from the manufacturer. Monitoring of ADA against TNF-α and VDZ was performed utilizing non-cross reactive ADA detection ELISA kits from Progenika Biopharma-Grifols according to the manufacturer's instruction. None of the sera showed detectable levels of ADA.

## Analysis of VDZ binding to exosomes

Exosomes isolated from a pool of blood donor's serum were incubated with increasing amount of VDZ (from 6.3 to 100 μg) for 2h at 4˚C on orbital shaker and purified again by Exoquick.

The levels of exosome-bound VDZ were investigated by western blot analysis. Purified exosomes were lysed in RIPA buffer and 20 μg of proteins were loaded for polyacrylamide native gel electrophoresis, then transferred to PVDF membranes for blocking and subsequent probing with primary antibodies against VDZ-HRP conjugated (1:1000, Progenika Biopharma-Grifols). Signals were detected using chemiluminescence method (SuperSignal™ West Femto substrate, Thermo Fischer Scientific).

## Analysis of exosome competition for binding to VDZ

Exosomes and CD4+ T cells purified from 1ml of blood donor serum were incubated alone or together with 60 μg of VDZ for 1h at 4˚C on orbital shaker, then cells were washed twice with PBS and lysed in RIPA buffer in ice for 30'. Protein extracted were separated by native electrophoresis and probed with primary antibodies against VDZ-HRP conjugated.

## MAdCAM adhesion assay

High-binding polystyrene 96-well plates were coated (100 μl/well) with 2.5 μg/mL of Recombinant Human MAdCAM-1 Fc Chimera (R&D system) diluted in PBS and incubated overnight at 37˚C. Nonspecific binding sites were blocked with 1% BSA in PBS w/$Ca^{2+}$/$Mg^{2+}$ for 1h at 37˚C. CD4+ T cells ($4x10^5$) were preincubated in binding buffer (1mM $MnCl_2$ and 1% BSA in PBS w/$Ca^{2+}$/$Mg^2$) with different concentration of VDZ (50 μg/mL, 5 μg/mL or 5 ng/mL) without or with $5x10^9$ exosomes for 15 min. Then, cells were added to MAdCAM-1-coated plate and incubated for 90 min at 37˚C. Adherent cells were stained with Hoechst dye and visualized by fluorescence microscopy (Leica DMI 6000B) coupled to a CCD camera (Leica DFC350FX).

## Statistical analysis

Data are expressed as the mean ± standard deviation. Unpaired t or Mann-Withney test was used to compare CTRL *vs* UC patients and anti- TNFα naïve *vs* anti-TNFα antagonist-experienced patients. $p < 0.05$ was considered statistically significant.

# Results

## Purification and characterization of serum exosomes from patients and controls

As illustrated in Fig 1A, the mean concentration of exosomes isolated from sera of UC patients was lower than in CTRL, even though the difference was not significant ($1.3x10^{11}$ ±$8.5x10^{10}$ versus $7.1x10^{10}$ ±$3.8x10^{10}$, p = 0.058). By contrast, exosomal concentration was greater in

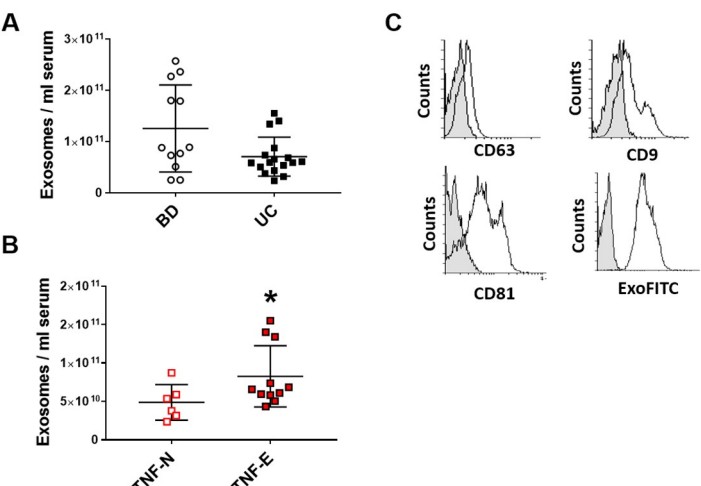

**Fig 1. Circulating exosomes isolation and characterization.** Exosomes were isolated from serum by Exoquick and quantified by Exocet. Panel A and B illustrate the mean concentration of exosomes in 12 blood donor (CTRL) and 17 UC patients, subdivided in anti- TNFα naïve (TNF-N) and anti-TNFα antagonist exposed (TNF-E). * p<0.05 compared to anti- TNFα naïve patients C) Exosomes were validated for the expression of exosomal markers by flow cytometry. Exosome-bound beads (white peak) were compared with beads alone (grey peak).

patients previously exposed to anti-TNFα agents compared to those naïve to biologics ($4.9x10^{11}\pm2.3x10^{10}$ versus $8.3x10^{10}\pm40x10^{10}$, p = 0.047) (Fig 1B).

Flow cytometry analyses demonstrated that exosomes isolated from UC patients and CTRL expressed the expected surface tetraspanins, such as the CD9, CD63 and CD81 and were positive for exosome stain Exo-FITC, a fluorescent dye that recognizes post-translational modifications on exosomal surface proteins (see Fig 1C as an example).

## Serum-derived exosomes expressed the $\alpha_4\beta_7$ integrin and were able to bound VDZ

The expression of exosomal marker CD9 (Fig 2B), T cells marker CD3 (Fig 2C) and monocytes marker CD14 (Fig 2D) was not different in exosomes isolated from UC patients compared with CTRL and from anti-TNFα-antagonist exposed patients compared to anti- TNFα naïve patients. Of note, serum-derived exosomes expressed the VDZ target α4β7 integrin (Fig 2E), but not its ligand, the MAdCAM-1 protein (Fig 2F). Exosomes from UC patients expressed more integrin $\alpha_4\beta_7$ than CTRL, however the difference was not statistically significant (16 ±12.4 versus 22.8±17.9, p = 0.097), whereas the expression of α4β7 integrin in exosomes of TNFα-antagonist exposed patients was greater than in anti- TNFα naïve patients (14.1 ±4.1 versus 27.6±20.8, p = 0.047).

To verify the hypothesis that circulating exosomes of UC patients expressing the $\alpha_4\beta_7$ integrin may carry VDZ in the follow-up, the amount of VDZ in serum, free and bound to exosomes, was measured by ELISA. Fig 3A shows that VDZ was measurable in serum both as free molecule and bound to exosomes. The expression of $\alpha_4\beta_7$ integrin and the presence of VDZ bound to UC exosomes was also confirmed by native immunoblotting (Fig 3B). As expected, exosomes isolated from serum of CTRL patient was negative for VDZ as they are not under therapy, while the signal was present in UC patients.

Considering the amount of VDZ bound by exosomes as a percentage of the total VDZ measured in serum (Fig 3C), we observed an increased sequestration of the medication in anti-TNFα -antagonist exposed compared to anti- TNFα naïve patients (61.5 ±26% versus 36.7

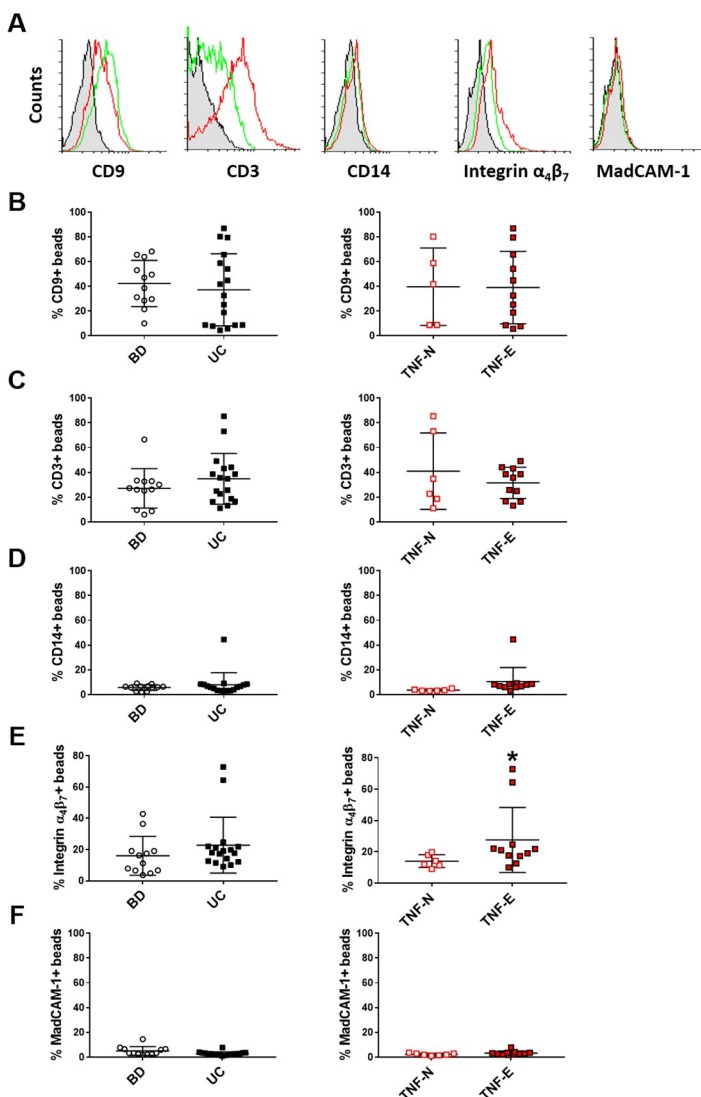

**Fig 2. Circulating exosomes express $\alpha_4\beta_7$ integrin.** (A) Purified vesicles isolated from serum of blood donor (CTRL) and UC patients subdivided in anti- TNF$\alpha$ naïve (TNF-N) and anti-TNF$\alpha$ antagonist exposed (TNF-E) were captured on antibody-coated beads and analyzed by flow cytometry. Representative plots of the FACS analyses for CTRL(green line) and UC-derived exosomes (red line) are shown. Graphs reported the expression levels (percentage of exosome-bound beads compared with beads alone) of CD9 (B), CD3 (C), CD14 (D), $\alpha_4\beta_7$ integrin (E) and MadCAM-1 (F) on exosome surface, * p<0.05 compared to anti- TNF$\alpha$ naïve patients.

±16.8%; p = 0.037). Finally, there was a correlation (trending towards significance) between the levels of $\alpha_4\beta_7$ integrin and the amounts of VDZ bound to exosomes in UC patients (Fig 3D).

Finally, to confirm that VDZ is bound to exosomes and does not precipitate as a contaminant, we compared exosomes isolated from a pool of UC patients sera (n = 5) by Exoquick with other two commercial kits (ExoEasy and qEV2). The concentration of exosomes varied in the samples isolated using the different methods and polymer-precipitation based method ensured the highest yield (Fig 4A). Immunoblotting analysis confirmed the presence of exosomal markers (CD9, TSG101 and flotillin-2) in all the exosomal preparations obtained with the different isolation methods (Fig 4B). The NTA analysis (Fig 4C) revealed that the size distribution of the isolated particles was similar for those obtained with ExoQuick (mean: 120±11.4

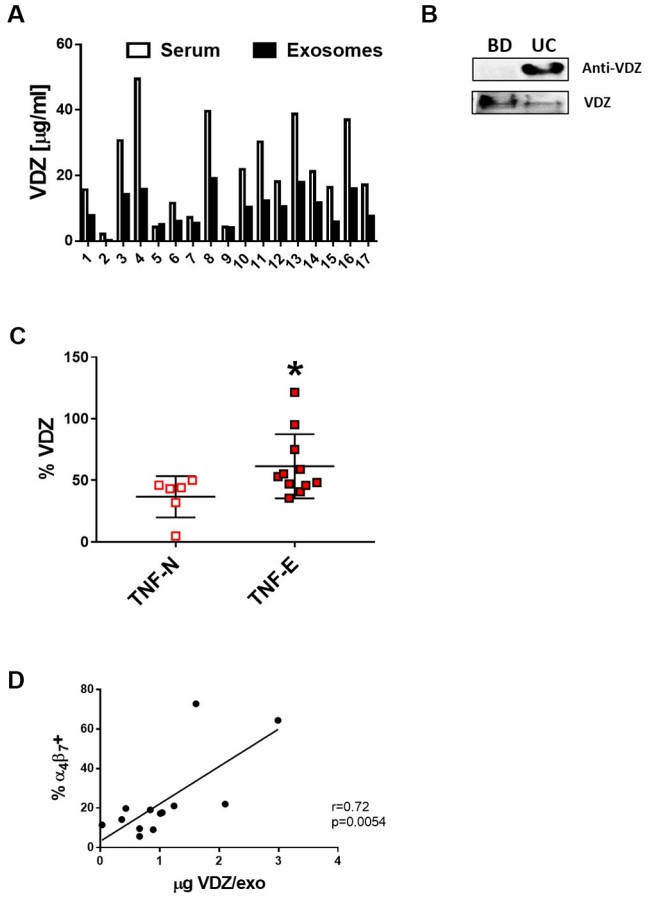

**Fig 3. VDZ is bound to exosomes in UC patients.** (A)Serum levels of VDZ free and bound to exosomes were quantified by Promonitor-VDZ ELISA. (B)The expression of $\alpha_4\beta_7$ integrin bound to VDZ was confirmed by native immunoblotting on exosomes lysate. (C)The sequestration of VDZ in exosomes was expressed as the percentage of VDZ bound to exosomes over the total level measured in serum. (D) Spearman correlation between the expression of $\alpha_4\beta_7$ integrin and the amount of VDZ bound to $10^{10}$ exosomes isolated from 12 UC patients.

nm, mode: 88.9±7.8 nm) and qEV2 (mean: 128.9±6.9 nm, mode: 96.5±6.7 nm). Notably, the size distribution profile of particles isolated by ExoEasy was significantly different compared to the other two isolation procedures, with more vesicles with higher size (mean: 174±8.5 nm, mode: 168.8±9.6 nm). We then analyzed the level of VDZ bound to exosomes by ELISA: the amount of VDZ sequestrated in $10^{10}$ exosomes was comparable among the three exosomal preparations (Fig 4D).

## Exosomes compete with T cells for binding to VDZ

To evaluate if VDZ is able to bind exosomes also from individuals never exposed to VDZ, we incubated CTRL exosomes with increasing amount of VDZ and then we analyzed exosomal lysate by native immunoblotting. The results demonstrated that exosomes were able to bind VDZ in a dose–dependent manner, depicting a parabolic curve to saturation (Fig 5A). As negative control, we also incubated VDZ alone with Exoquick in order to exclude its precipitation as a contaminant during the isolation procedure.

In order to evaluate if exosomes compete with T cells for VDZ binding, we incubated purified CD4+ T cells with VDZ alone or with a fixed amount of exosomes; the levels of VDZ

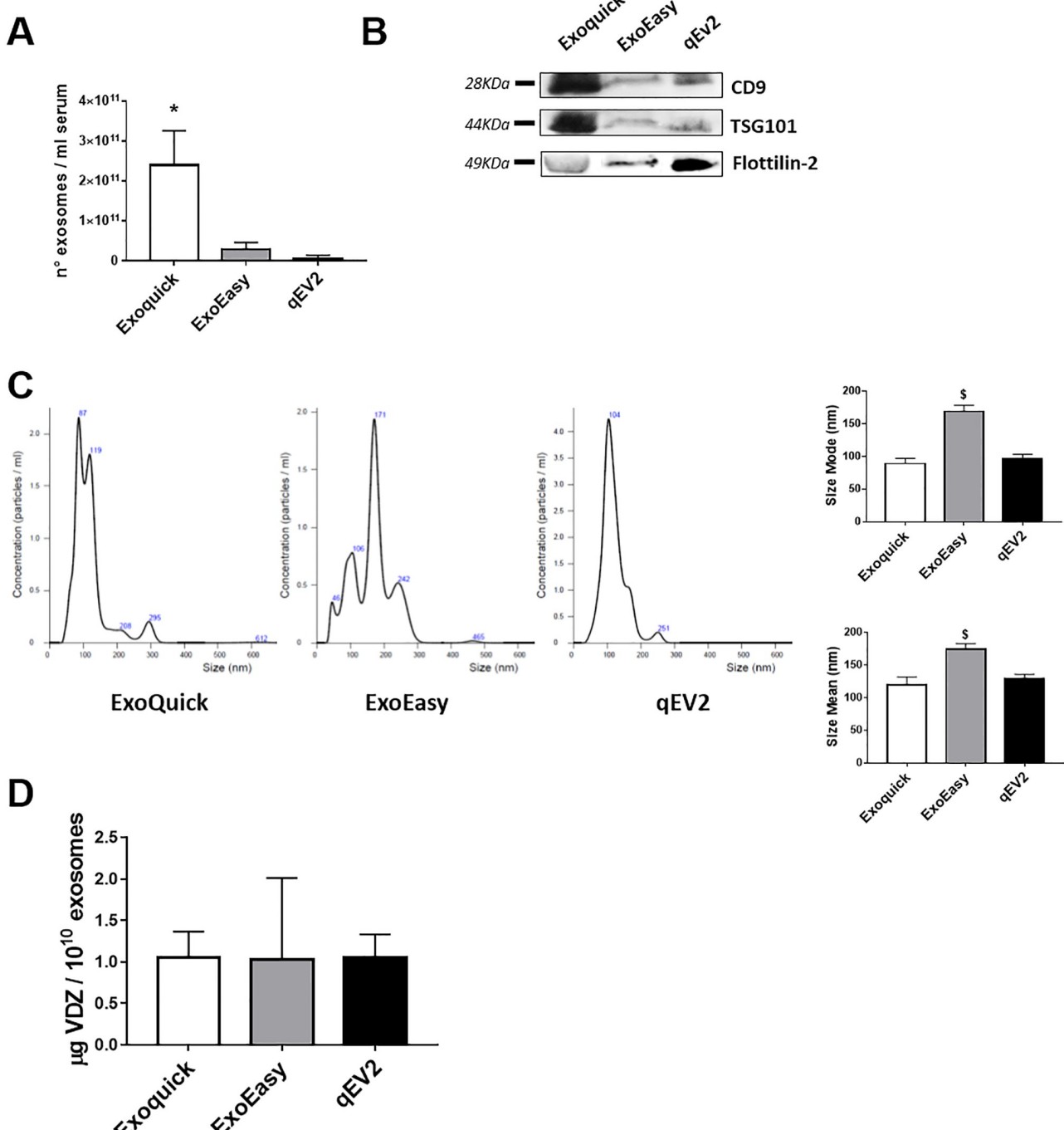

**Fig 4. Comparison of three different methods of exosome isolation.** Exosomes were isolated from a pool of patient's sera by polymer precipitation (Exoquick), membrane-affinity (ExoEasy) and size-exclusion column (qEV2) methods. Vesicles were quantified by Exocet (A), validated for the expression of exosomal markers by immunoblotting (B) and analysed for size distribution by NTA (C). The levels of VDZ bound to exosomes were quantified by ELISA. Data are shown as mean ± SD. * $p < 0.05$ compared to exosomes isolated by ExoEasy and qEV2; $ $p < 0.05$ compared to exosomes isolated by ExoQuick and qEV2.

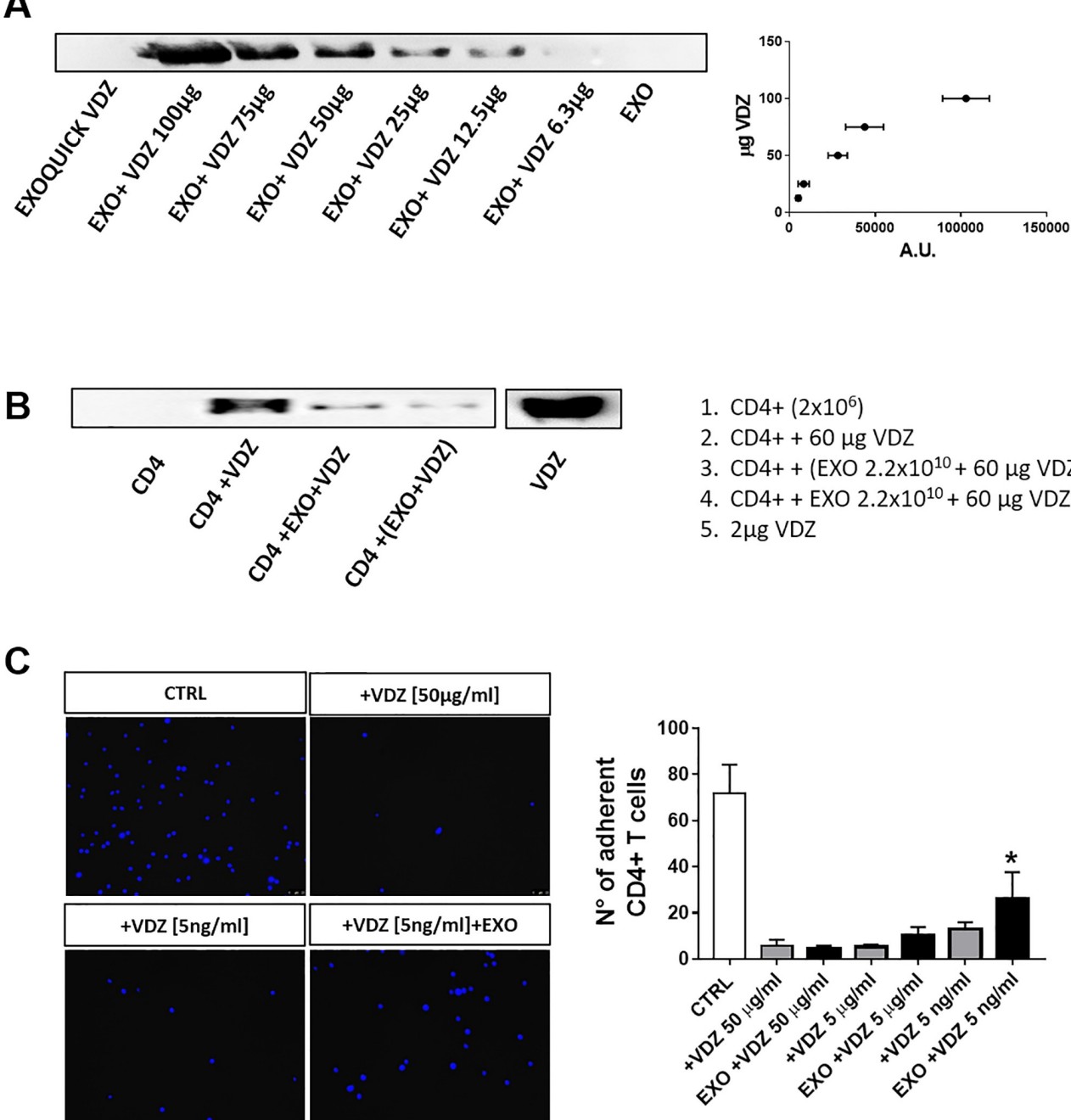

**Fig 5. Exosomes compete with T cells for binding to VDZ.** VDZ bound to exosomes (A) or to CD4+ T cells (B) were evaluated by native immunoblotting. One representative blot is shown of three independent experiments. (C) CD4+ T cells were incubated with VDZ with or without exosomes and added to MAdCAM-1-coated plate. Representative microscope images of adherent cells stained with Hoechst dye are shown. Data are shown as mean ± SD. * p<0.05 compared to adherent cells treated with 5 ng/ml VDZ alone.

bound to cells were then measured by native immunoblotting. We found that the amount of VDZ bound to cells was impaired in the presence of exosomes, either if exosomes were pre-incubated with VDZ or given with VDZ simultaneously to CD4+ T cells (Fig 5B). We next evaluated the capacity of exosomes to interfere with the adhesion of VDZ-treated CD4+ T cells

(Fig 5C). As expected, VDZ significantly suppressed the adhesion of CD4+ T cells to MAd-CAM-1 coated plates. Of note, the number of adherent cells were increased in the presence of exosomes. Hence, the data suggest that exosomes are able to sequestrate VDZ, decreasing the amount available for binding to T cells.

## Discussion

Recent studies have revealed that exosome crosstalk mechanisms may influence major IBD-related pathways, such as immune responses, barrier function and intestinal flora [21]. In our study, we propose a completely new role of exosomes in IBDs, whereby vesicles expressing the $\alpha_4\beta_7$ integrin might bind VDZ and interfere with drug bioavailability and efficacy.

As with other biologic therapies, a variable proportion of patients do not respond to VDZ at induction or they might lose response over time. Of note, the immunogenicity of VDZ is very low if compared to that of anti-TNF-agents, since the development of anti-VDZ antibodies seems to occur rarely (in 2–4% of cases) [22–24] and therefore does not seem to affect the clinical efficacy of the drug [25]. In fact, anti-VDZ antibodies usually appear after the first infusion, but they are transient [23] and do not correlated with clinical outcomes [22]. In patients who lose response to VDZ over time, the most common management strategy is to reduce the therapeutic interval between VDZ infusions, from 8 to 4 weeks, in order to raise the drug concentration until the next medication dose [26]. Although optimal VDZ trough levels during induction and maintenance have been shown to be associated with endoscopic and clinical outcomes [27], some patients do not respond to this therapy even when adequate trough levels are reached, suggesting that, in these cases, "therapeutic" drug concentrations may be not sufficient to predict clinical response [25]. Therefore, new strategies to predict therapeutic response might be needed.

Our data demonstrate that circulating exosomes express the VDZ target $\alpha_4\beta_7$ integrin and bind VDZ, decreasing its bioavailability. Of note, we confirmed the presence of VDZ bound to the vesicles using three different techniques, based on different isolation methods, thus avoiding possible contamination related to a single method.

When VDZ binds to exosomes, it may interfere with their regulatory activity on T cells trafficking into the gut. Exosomes secreted from gut-tropic memory/effector T cells act as a negative regulator that can adjust the levels of gut-specific lymphocyte homing by suppressing MAdCAM-1 expression in the small intestine [28]. Moreover, our data suggest that exosomes could be involved in mediating VDZ resistance by sequestration of therapeutic molecules into vesicles. In accordance with our hypothesis, it has been already reported that exosomes secreted by breast cancer cells carrying HER2 antigen, the target of Trastuzumab, contribute to drug sequestration and compromise its efficacy [18]. Similarly, exosomes released from B-cell lymphomas carrying the CD20 target antigen, may act as decoy targets upon Rituximab exposure, allowing lymphoma cells to escape from humoral immunotherapy [29].

In general, IBD patients previously treated with biologic agents are less likely to respond to a second biologic drug [30]. In our study, we observed that the serum exosomal concentration and the expression of $\alpha_4\beta_7$ integrin on vesicles surface was greater in anti-TNF$\alpha$-antagonist exposed patients compared to anti- TNF$\alpha$ naïve patients. Accordingly, we reported an increased exosomal sequestration of VDZ in anti-TNF$\alpha$-antagonist exposed patients, suggesting that previous biologic therapy may have altered the sequestration capacity of circulating exosomes, thus reducing the efficacy of VDZ in patients who failed anti-TNF$\alpha$ agents. In agreement, a trend of significant correlation was found between exosomal integrin expression and drug sequestration.

Our results could explain the low and slow response rate of VDZ in patients with a large inflammatory burden whereby the efficacy of the drug could be completely or partially

compromised by the increase in $\alpha_4\beta_7$ integrin expression in exosomes, which could bind and inactivate the drug. However, the initial combination of VDZ and steroids or calcineurin inhibitors could reduce exosome formation, decrease drug sequestration and increase VDZ efficacy [31, 32].

There are several limitations to the present study. The retrospective design and the very low number of patients involved do not allow us to establish a significative correlation between our findings and the clinical and laboratory behavior of the disease. Moreover, the single blood sample obtained at different time points in individual patients might not entirely reflect the disease behavior and the drug efficacy over time.

In conclusion, this is the first report demonstrating that exosomes bind VDZ and that such binding may interfere with its pharmacokinetics. Additional, larger studies will assess whether VDZ sequestration in exosomes may correlate with clinical and endoscopic response and whether it is possible to interfere with such binding to increase drug bioavailability.

## Acknowledgments

We thank Prof. Dario Sorrentino (IBD Center, Division of Gastroenterology, Virginia Tech Carilion School of Medicine, Roanoke, Virginia) or its valuable support in proofreading and editing of this work.

## Author Contributions

**Conceptualization:** Rossana Domenis, Marco Marino, Martina Fabris.

**Formal analysis:** Adriana Cifù.

**Funding acquisition:** Francesco Curcio.

**Investigation:** Rossana Domenis, Adriana Cifù, Giulia Scardino.

**Methodology:** Rossana Domenis, Adriana Cifù, Giulia Scardino.

**Resources:** Francesco Curcio.

**Supervision:** Rossana Domenis, Marco Marino, Francesco Curcio, Martina Fabris.

**Writing – original draft:** Rossana Domenis, Marco Marino, Francesco Curcio, Martina Fabris.

**Writing – review & editing:** Rossana Domenis, Marco Marino, Francesco Curcio, Martina Fabris.

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
