## [Decision Letter · Decision Letter 0]

10 Sep 2020

PONE-D-20-23481

Circulating exosomes express α4β7 integrin and compete with CD4+ T cells for the binding to Vedolizumab

PLOS ONE

Dear Dr. Fabris,

Thank you for submitting your manuscript to PLOS ONE. After careful consideration, we feel that it has merit but does not fully meet PLOS ONE’s publication criteria as it currently stands. Therefore, we invite you to submit a revised version of the manuscript that addresses the points raised during the review process.

As you may note, while both reviewers thought the study was important, both of them had both conceptual/interpretational and technical concerns. Please respond to each of the issues raised in a satisfactory manner.

We look forward to receiving your revised manuscript.

Kind regards,

Aftab A. Ansari, PhD

Academic Editor

PLOS ONE

Journal Requirements:

"This work was supported by Grant Interreg ARTE (J22F170001005) and partially supported by an unrestricted grant from VivaBioCell S.p.A.

The funders had no role in study design, data collection and analysis, decision to publish, or preparation of the manuscript. "

We note that you received funding from a commercial source: VivaBioCell S.p.A.

Reviewers' comments:

Reviewer's Responses to Questions

**Comments to the Author**

1. Is the manuscript technically sound, and do the data support the conclusions?

Reviewer #1: Partly

Reviewer #2: Yes

2. Has the statistical analysis been performed appropriately and rigorously? 

Reviewer #1: I Don't Know

Reviewer #2: Yes

3. Have the authors made all data underlying the findings in their manuscript fully available?

Reviewer #1: No

Reviewer #2: No

4. Is the manuscript presented in an intelligible fashion and written in standard English?

Reviewer #1: Yes

Reviewer #2: Yes

5. Review Comments to the Author

Reviewer #1: The study reported by Domenis et al reports a proof of the existence of a mechanisms that can consume Vedolizumab in vivo, via shedding of exosomes from T cells. This as pointed out in the manuscript is not unheard of as cells release exosomes bearing some but not all their membrane proteins. Of note the circulating exosomes isolated appear predominantly released from T cells based on CD3 expression. The data goes to great lengths to demonstrate the removal of vedolizumab by exosomes, and its ability to affect T cell binding to MAdCAM-1 in vitro. There is also ex vivo demonstration demonstration of a4B7 expression on plasma exosomes, but the study did not include additional data to show that the therapy failure was actually due to exosomes consuming the mAb. There are several concerns about the manuscript in its current form:

1) Study population: The number of patients is very small and heterogeneous. Individual ages are not listed and that leaves the reader to wonder whether select associations may be possible with exosome recovery etc. There was no scoring of UC included. Since the conclusions suggest that exosomes might have contributed to the failure of the Vedolizumab therapy at the very least, ADA should have been checked against the same Mab. It is quite possible that patients failing anti-TNF-a may have developed ADA that either cross reacts with, or at least cross primes the patient for Vedolizumab. As a more minor point, the text mentions “naïve patients” several times which is confusing. Specify “anti-TNF-a naïve”. Next, the table of patients lists 6 patients in the TNF-a naïve group as having undergone anti-TNF agent therapy??? Which is it?

2) Considering that most exosomes are born from T cells, It would have been quite informative also to have flow cytometry conducted on CD4 and CD8 T cells for activation markers including a4B7 and test whether these expression markers correlate with the level of a4B7 expressed by exosomes.

3) Minor point: the controls are “BD” which is very close to IBD, perhaps an unfortunate choice for the reader.

4) Figure 1A,B: There is quite a bit of variability in the amount of exosomes isolated. Is the isolation method truly quantitative? Controls have higher non significant numbers of exosomes than all 17 UC patients together. But then, the anti-TNF-exposed patient samples are statistically higher though that is driven only by 3 values. This was not really addressed and one wonders how reliable these values might be.

5) Figure 2: Same concern for the a4B7 exposure in samples from only 2 anti-TNF-a exposed patients. This could be due to limited numbers since there is no difference with controls. The same applies to Fig 3C.

6) Figure 3B is confusing as presented. Exosomes from control patient cannot contain VDZ since they did not under such treatment, this needs to be specified in the figure legend.

7) Figure 4A and B raise new questions regarding the various exosome purification methods. There are quantitative but also qualitative differences in the isolation of exosomes that are not commented upon. The qEV2 method, appear far less efficient but then again these exosomes express markedly more Flottilin than others. What does that mean? Different subtypes of exosomes isolated? At the very least these should be characterized for the CD3 and CD14 expression. The scale of Fig 4A needs to be changed to present the actual numbers purified even via the qEV2 method.

8) Figure 5: How many replicates were performed for the binding curve? There were no error bars. The picture in 5C are also very dark and difficult to read. If possible provide lighter ones.

Reviewer #2: In this manuscript Domenis et al., made attempts to characterize exosomes isolated from serum of ulcerative colitis patients treated with VDZ and address the possibility of circulating exosomes bind VDZ and interfere with its therapeutic efficacy.

However, It is well established in the field that EVs coming out of a4b7 expressing cells carry the a4b7 on their surface and it is obvious that after infusion of mAB against a4b7 for treatment of IBD, a fraction of the mab is going to binds to these EVs.

The major question is that, how this EV bound Mabs effects the therapeutic concentration and bioavailability of the mab? And how that phenomenon going to reduce the masking of a4b7 within the a4b7 expressing gut homing lymphocytes and is responsible for therapeutic resistance to VDZ? was not addressed mechanistically.

Although the authors demonstrated that a4b7 expressing EVs bound VDZ in serum, but how that compromise the masking of a4b7 in gut homing lymphocyte remain unanswered?

The most relevant experiment to prove that, binding of VDZ to EVs reduce the effective therapeutic concentration of VDZ and is responsible to VDZ resistance, is to simply show the % masking of a4b7 that is achieved within the a4b7+ gut homing lymphocytes from the clinical sample population used in the study? Therefore, it will great idea that authors can performing this experiment to establish the hypothesis presented in the manuscript that EV bound VDZ based lowering of VDZ bioavailability is responsible for therapeutic resistance of VDZ?' will be great contribution to the VDZ field.

Otherwise hypothesis and idea is good but important experiment is missing?

6. PLOS authors have the option to publish the peer review history of their article (what does this mean?). If published, this will include your full peer review and any attached files.

Reviewer #1: No

Reviewer #2: No

---

## [Author Response · Author response to Decision Letter 0]

28 Oct 2020

Reviewer #1

The study reported by Domenis et al reports a proof of the existence of a mechanisms that can consume Vedolizumab in vivo, via shedding of exosomes from T cells. This as pointed out in the manuscript is not unheard of as cells release exosomes bearing some but not all their membrane proteins. Of note the circulating exosomes isolated appear predominantly released from T cells based on CD3 expression. The data goes to great lengths to demonstrate the removal of vedolizumab by exosomes, and its ability to affect T cell binding to MAdCAM-1 in vitro. There is also ex vivo demonstration of a4B7 expression on plasma exosomes, but the study did not include additional data to show that the therapy failure was actually due to exosomes consuming the mAb. There are several concerns about the manuscript in its current form:

1) Study population: The number of patients is very small and heterogeneous. Individual ages are not listed and that leaves the reader to wonder whether select associations may be possible with exosome recovery etc. There was no scoring of UC included. 

As suggested, demographic and disease characteristics, including Mayo score, of the each patient were listed in two new table, in which patients were subdivided in anti-TNFα naïve and anti-TNFα antagonist exposed.

Since the conclusions suggest that exosomes might have contributed to the failure of the Vedolizumab therapy at the very least, ADA should have been checked against the same Mab. It is quite possible that patients failing anti-TNF-a may have developed ADA that either cross reacts with, or at least cross primes the patient for Vedolizumab. 

Anti-TNF therapies and Vedolizumab act against different targets that do not cross-react. There is no evidence that ADA developed against anti-TNF agents may cross-react or prime patients when treated with Vedolizumab. The ELISA kit we used to test ADA against anti-TNF agents are specific for these therapeutic agents and the ELISA we used to test ADA against Vedolizumab are specific for this drugs and gave negative results in patients who previously developed ADA against anti-TNF agents. But, it is well recognized that patients developing ADA against a first biologic agent are at risk to develop an immune response against other biological agents in general. Perhaps, with the sentence “cross prime”, the Reviewer intended that a patients is prone to develop immunereactivity against VDZ, but exosomes eventually implicated in anti-TNF agents sequestration acts differently compared to those binding VDZ since the target on exosomes is different (TNF versus α4β7) and the ADA specifically directed against a different drug. 

As a more minor point, the text mentions “naïve patients” several times which is confusing. Specify “anti-TNF-a naïve”. 

We specify “anti-TNFα naïve patients” throughout the text, as suggested by reviewer.

Next, the table of patients lists 6 patients in the TNF-a naïve group as having undergone anti-TNF agent therapy??? Which is it?

We apologized for the mistake.

2) Considering that most exosomes are born from T cells, It would have been quite informative also to have flow cytometry conducted on CD4 and CD8 T cells for activation markers including a4B7 and test whether these expression markers correlate with the level of a4B7 expressed by exosomes.

The suggestion of the reviewer is very interesting, but we cannot do this experiment since we used the samples stored for this study to test exosomes and we do not have T cells frozen available for this experiment, but we take into consideration this suggestion for the next studies.

3) Minor point: the controls are “BD” which is very close to IBD, perhaps an unfortunate choice for the reader.

We agree with reviewer and we replace “BD” with “CTRL”.

4) Figure 1A,B: There is quite a bit of variability in the amount of exosomes isolated. Is the isolation method truly quantitative? Controls have higher non significant numbers of exosomes than all 17 UC patients together. But then, the anti-TNF-exposed patient samples are statistically higher though that is driven only by 3 values. This was not really addressed and one wonders how reliable these values might be.

5) Figure 2: Same concern for the a4B7 exposure in samples from only 2 anti-TNF-a exposed patients. This could be due to limited numbers since there is no difference with controls. The same applies to Fig 3C.

We agree that the difference between TNF-N and TNF-E in exosomal concentration and integrin expression was near the limit of significance (p=0.047), but was supported by the significant increased sequestration of VDZ in TNF-E compared to TNF-N patients (61.5 ± 26% versus 36.7 ±16.8%; p=0.037). In other words, the higher number of exosomes combined with a higher expression of integrin in TNF-E compare to TNF-N patients results in a greater drug sequestration. Obviously, we believe that exosomes may be only one of the causes of the failure or loss of response to therapy. 

We agree with the reviewer that the number of clinical cases is low, but we are already started a second study with a wider patients population aimed to confirm our observations. 

6) Figure 3B is confusing as presented. Exosomes from control patient cannot contain VDZ since they did not under such treatment, this needs to be specified in the figure legend.

We thank the reviewer for this observation and we have modified the text specifying that “as expected, exosomes isolated from serum of CTRL patient was negative for VDZ as they are not under therapy, while the signal was present in UC patients.”

7) Figure 4A and B raise new questions regarding the various exosome purification methods. There are quantitative but also qualitative differences in the isolation of exosomes that are not commented upon. The qEV2 method, appear far less efficient but then again these exosomes express markedly more Flottilin than others. What does that mean? Different subtypes of exosomes isolated? At the very least these should be characterized for the CD3 and CD14 expression. The scale of Fig 4A needs to be changed to present the actual numbers purified even via the qEV2 method.

Since the discovery of exosomes, a gold standard method for their isolation has not yet been identified and the technique used by different research groups was always a compromise between purity and yield. It is clearly recognized that the different exosomal enrichment methods yield different subpopulations, which could be can be distinguished on the basis of size and expression of the exosomal markers. For example, large EXOs secreted from mesenchymal stem cells were found to be enriched in CD63 and flotillin-1, whereas small EXOs were enriched in ALIX and TSG101 (Willis GR et al. Toward exosome-based therapeutics: isolation, heterogeneity, and fit-for-purpose potency. Front Cardiovasc Med (2017) 4:63.10.3389/fcvm.2017.00063). 

For these reason, the International Society for Extracellular Vesicles (ISEV) specifies that there is no single optimal separation method and the choice must be made based on the downstream applications and scientific question. In addition, it recommends to validate data by testing different isolation methods (Lötvall et al. (2014) Minimal experimental requirements for definition of extracellular vesicles and their functions: a position statement from the International Society for Extracellular Vesicles, Journal of Extracellular Vesicles, 3:1, 26913, DOI: 10.3402/jev.v3.26913). 

Accordingly, we did such comparative measures to show that, regardless the isolation technique we used , VDZ binds to the exosomes and the quantity does not change. Consequently, the data reported were not artifacts due to the specific isolation process or the enrichment in a specific exosomal subpopulation. 

To better represent the actual number of exosomes obtained by the qEV2 isolation method, we modified the scale of the graph in figure 4A as suggested.

8) Figure 5: How many replicates were performed for the binding curve? There were no error bars. The picture in 5C are also very dark and difficult to read. If possible provide lighter ones.

We specify that one representative blot is shown of three independent experiments in figure legend and we insert the error bars in the graph. 

As suggested, we insert lighter and bigger images in picture 5C.

Reviewer #2

In this manuscript Domenis et al., made attempts to characterize exosomes isolated from serum of ulcerative colitis patients treated with VDZ and address the possibility of circulating exosomes bind VDZ and interfere with its therapeutic efficacy.

However, It is well established in the field that EVs coming out of a4b7 expressing cells carry the a4b7 on their surface and it is obvious that after infusion of mAB against a4b7 for treatment of IBD, a fraction of the mab is going to binds to these EVs.

The major question is that, how this EV bound Mabs effects the therapeutic concentration and bioavailability of the mab? And how that phenomenon going to reduce the masking of a4b7 within the a4b7 expressing gut homing lymphocytes and is responsible for therapeutic resistance to VDZ? was not addressed mechanistically.

Although the authors demonstrated that a4b7 expressing EVs bound VDZ in serum, but how that compromise the masking of a4b7 in gut homing lymphocyte remain unanswered?

The most relevant experiment to prove that, binding of VDZ to EVs reduce the effective therapeutic concentration of VDZ and is responsible to VDZ resistance, is to simply show the % masking of a4b7 that is achieved within the a4b7+ gut homing lymphocytes from the clinical sample population used in the study? Therefore, it will great idea that authors can performing this experiment to establish the hypothesis presented in the manuscript that EV bound VDZ based lowering of VDZ bioavailability is responsible for therapeutic resistance of VDZ?' will be great contribution to the VDZ field.

Otherwise hypothesis and idea is good but important experiment is missing?

We greatly appreciated the suggestion of the reviewer and we believe that the proposed experiment may be useful to further improve our data. 

Nevertheless, the aim and originality of this study was to demonstrate that exosomes expressing α4β7 on their surface can bind VDZ and thus potentially interfere with drug availability and possibly, therapeutic efficacy. 

To our knowledge, with the exception of the study published by Ciravolo on Trastuzumab, our study is the only one to highlight that exosomes can reduce drug availability. We therefore agree that the number of patients in the present study is low, but sufficient to demonstrate our aim. 

We are already working on a much larger series of patients who will be tested either at induction and during the maintenance phases and VDZ sequestration rate by exosomes will be correlated with the clinical response.

---

## [Editor Report · Decision Letter 1]

2 Nov 2020

Circulating exosomes express α4β7 integrin and compete with CD4+ T cells for the binding to Vedolizumab

PONE-D-20-23481R1

Dear Dr. Fabris,

We’re pleased to inform you that your manuscript has been judged scientifically suitable for publication and will be formally accepted for publication once it meets all outstanding technical requirements and you have added the sentence I have recommended within the text, if this sentence is accurate.

Kind regards,

Aftab A. Ansari, PhD

Academic Editor

PLOS ONE

Additional Editor Comments (optional):

I would suggest that you add a sentence at the end of lines 161-162 such as "Monitoring of ADA against TNF-alpha and VDZ was performed utilizing non-cross reactive ADA detection ELISA kits from Progenica Biopharma-Grisfol according to the manufacturer's instruction. None of the sera showed detectable levels of ADA.
---

## [Editor Report · Acceptance letter]

4 Nov 2020

PONE-D-20-23481R1 

Circulating exosomes express α_4_β_7_ integrin and compete with CD4+ T cells for the binding to Vedolizumab 

Dear Dr. Fabris:

I'm pleased to inform you that your manuscript has been deemed suitable for publication in PLOS ONE. Congratulations! Your manuscript is now with our production department. 

Kind regards, 

on behalf of

Dr. Aftab A. Ansari 

Academic Editor

PLOS ONE